# Comparative Study of the Antibacterial, Biodegradable, and Biocompatibility Properties of Composite and Bi-Layer Films of Chitosan/Gelatin Coated with Silver Particles

**DOI:** 10.3390/ma16083000

**Published:** 2023-04-10

**Authors:** Laura-Elizabeth Valencia-Gómez, Hortensia Reyes-Blas, Juan-Francisco Hernández-Paz, Claudia-Alejandra Rodríguez-González, Imelda Olivas-Armendáriz

**Affiliations:** Instituto de Ingeniería y Tecnología, Universidad Autónoma de Ciudad Juárez, Ave. Del Charro #610 Norte, Col. Partido Romero, Cd. Juárez 32320, Mexicohortensia.reyes@uacj.mx (H.R.-B.);

**Keywords:** biodegradable polymers, biocompatible polymers, coatings, chitosan, gelatin

## Abstract

The dressings are materials that can improve the wound-healing process in patients with medical issues. Polymeric films are frequently used as dressings with multiple biological properties. Chitosan and gelatin are the most used polymers in tissue regeneration processes. There are usually several configurations of films for dressings, among which the composite (mixture of two or more materials) and layered ones stand out (layers). This study analyzed the antibacterial, degradable, and biocompatible properties of chitosan and gelatin films in 2 configurations, composite and bilayer, composite. In addition, a silver coating was added to enhance the antibacterial properties of both configurations. After the study, it was found that the bilayer films have a higher antibacterial activity than the composite films, having inhibition halos between 23% and 78% in Gram-negative bacteria. In addition, the bilayer films increased the fibroblast cell proliferation process, reaching up to 192% cell viability after 48 h of incubation. On the other hand, composite films have greater stability since they are thicker, with 276 µm, 243.8 µm, and 239 µm compared to 236 µm, 233 µm, and 219 µm thick for bilayer films; and a low degradation rate compared to bilayer films.

## 1. Introduction

The wound healing process has multi-factorial physiological activities and can be promoted with wound treatment or dressings. There currently exists various dressings for healing skin, among which are: (a) Traditional dressings that are used in the first stage of therapy to stop bleeding and create a cover between the wound and environment; (b) Biomaterials-based dressings, biomimetic polymers with collagen-type structures; (c) Artificial dressings, that are the most effective materials in wound healing and have a longer shelf life [1].

Carbohydrates and proteins are the most used macromolecules for elaborated film dressings with biocompatible and antimicrobial properties. Chitosan and gelatin are the most common biopolymers for this application. On the one hand, chitosan is a versatile, functional polysaccharide comprising b-(1,4)-linked N-acetyl-D-glucosamine and D-glucosamine monomers. However, one of its drawbacks is that it is insoluble in water and soluble in acid solutions, because of the positive charge amino group on the carbon 2 of the glucosamine at pH below 6 [2]. Due to its antimicrobial, biocompatible, and biodegradable properties, chitosan is considered a promising material for regenerative applications, such as wound healing.

Furthermore, this material also has properties associated with wound treatment, such as cell adhesion, antifungal, and oxygen permeability, among others [3]. On the other hand, gelatin, a collagen derivative, approximates collagen’s biological structure in autochthonous tissues of the extracellular matrix; in addition, it preserves its natural cell binding characteristics, such as arginine-glycine-aspartic motifs, which favor cellular responses, such as adhesion, proliferation, migration, and differentiation. Gelatin and other biopolymers such as alginate, collagen, and hyaluronic acid have become a research focus for evaluation in various biomedical applications [4].

Several investigations have used chitosan and gelatin to elaborate composite films to improve polymer complexes’ mechanical, physical, and biological properties [1,5]. These composite films can be created of polyelectrolyte complexes through electrostatic interactions between the ammonium groups of the chitosan chain and the negatively charged side-chain groups of the gelatin [1].

Silver particles are used in skin lesion dressings, given their low level of toxicity to human cells, easy access, and strong antimicrobial effect [6]. Dressings with antimicrobial properties protect the wound from infection and eliminate pain. Silver is an antimicrobial agent that has shown activity against more common pathogens such as *Staphylococcus aureus*, *Escherichia coli*, and *Pseudomonas* spp [7]. In addition, this material has previously been used as an antimicrobial agent for biomedical applications, adding it using the sputtering method, which has been reported as a technique that allows depositing a thin layer of silver, which, depending on its deposition time, may remain with the characteristic antimicrobial character of this material [8].

Various recent investigations have been carried out on manufacturing chitosan-based bilayer films. However, these studies are focused on the preparation of food packaging, for example, chitosan–poly (vinyl alcohol) bilayer films for strawberry packaging [9] and modified chitosan/chitosan nanoparticle and polyvinyl alcohol/starch bilayer films [10]. However, there is limited research on chitosan-based bilayer films for wound healing applications. For example, chitosan/konjac glucomannan bilayer film as a wound dressing, where it is observed that bilayer films have low cytotoxicity and inhibit microbial penetration [11]. Further, in 2019, gelatin/chitosan bilayer hydrofilms with crosslinking agents were studied, showing good properties for wound dressing use [4]. Nevertheless, chitosan/gelatin films coated Ag have not been studied for wound healing applications.

The objective of this investigation is to perform a comparative study of the biocompatible, antibacterial, and biodegradable properties of the composite and bi-layer films, elaborated with chitosan and gelatin and coated with silver particles, to determine the most efficient configuration for dressings on wound healing applications, through the study of cell viability in vitro, enzymatic degradation in vitro, the halo of inhibition, and contact angle of the elaborated films (Figure 1). This comparative study will help to understand the effect of the configuration of materials on the properties of chitosan/gelatin films. In addition, this research will provide more information on the use of bilayer films in wound care applications due to the limited current information on this type of film for medical applications.

## 2. Materials and Methods

### 2.1. Materials

The materials and solvents used to elaborate the films were chitosan from shrimp shells ≥ 75% deacetylated (Sigma-Aldrich, Monterrey, México C3646), gelatin from bovine skin (Sigma-Aldrich, G9382), acetic acid (T.Baker, Miami, FL, USA, 9508-05). For the characterizations and tests, the solvents were Dimethyl sulfoxide (DMSO) (Sigma-Aldrich, D4540), 3-(4,5-Dimethyl-2-thiazolyl)-2,5-diphenyl-2H-tetrazolium Bromide or MTT (Sigma-Aldrich, 475989), 1x DMEM culture medium (Gibco, Shanghai, China), 10% Fetal Bovine Serum (ATCC), 2% penicillin (Gibco), Accutase enzyme (Sigma-Aldrich), Soy Agar (Difco, Tucker, GA, USA, 236950), Nutrient Agar (Difco, 21300), and lysozyme enzyme (Fisher Scientific, Monterrey, México, BP535-1).

### 2.2. Composite Films and Bi-Layer Films Preparation

The composite films were prepared using a modified methodology reported by Valencia-Gomez, in 2022 [5]. Three different gelatin solutions were prepared with different concentrations, 2%, 2.5%, and 3%, dissolving the gelatin in distilled water, for 20 min in constant agitation and at room temperature. Then, a chitosan solution of 2% was elaborated with 2 g of the polymer and 100 mL of acetic acid solution of 1%, obtaining the 2%C, 2.5%C, and 3%C films (Table 1).

For the bi-layer film, 25 mL of 2%, 2.5%, and 3% of gelatin solutions were prepared. Each gelatin solution was poured into a petri dish and refrigerated at 2 °C for gelation. After, 25 mL of chitosan solution of 2% was also poured into the gelation gelatin solution. Finally, the films were coated for 15 min by sputtering technique on the chitosan layer, using a Ted Pella 99.9% silver (Ag) target (Table 1).

### 2.3. Characterizations

#### 2.3.1. Fourier Transform Infrared Spectroscopy (FTIR)

The FT-IR analysis was used to determine the functional groups of the composite and bi-layer films. The analysis was performed with a Nicolet 6700 spectrometer in the region of 600–3500 cm^−1^.

#### 2.3.2. Thickness: Scanning Electronic Microscopy

The thickness of the composite and bilayer films were studied using a scanning electron microscope (SU5000, Hitachi, Tokyo, Japan). The films were cut into pieces of 1 cm^2^. Subsequently, they were placed one by one with conductive carbon tape in a sample holder. Finally, the images were captured using the low-vacuum mode with an accelerating voltage of 10 kV, a magnification of 250×, and a pressure of 30 Pa.

#### 2.3.3. Surface Hydrophobicity Study

The hydrophilic character of the composite and bilayer films was evaluated by measuring the contact angle by the deposition of two drops of distilled water on the surface of the films. One drop on the surface of the silver and another drop on the surface of the polymer of each sample, at room temperature of 25 °C, using a goniometer (KRUSS, DSA30). The sample dimensions were 1 × 5 cm.

#### 2.3.4. Degradation In Vitro Assay

All films were cut into circles of 1 cm in diameter and weighed before summing on 1 mL 1X PBS solution with 0.02% sodium azide (NaN_3_) and 5µg/mL lysozyme enzyme. Subsequently, the samples were incubated for 1, 2, and 3 days at 37 °C. Finally, each sample was washed with distilled water and dried at room temperature. The percentage of weight loss (%WL) was calculated with the following equation:
(1)$$\%\mathrm{WL}=\frac{\mathrm{Wi}-\mathrm{Wf}}{\mathrm{Wi}}\times 100\%$$
where:Wf = final weightWi = initial weight

#### 2.3.5. Antibacterial Activity Assessment

The antibacterial activity was evaluated by agar diffusion method according to Pareda and cols [2], using the *Escherichia coli* (*E. coli*) (ATCC 11229TM) as standard for Gram-negative bacteria and *Staphylococcus aureus* (*S. aureus*) (ATCC 6538TM) as standard for Gram-positive bacteria. All films were cut into 1 cm of diameter pieces and placed on the plates seeded with *E. coli* and *S. aureus*, respectively. Then, the seeded plates were incubated at 37 °C for 24 h. After incubation, the growth inhibition zone was measured with the help of the ImageJ 2 software.

#### 2.3.6. In Vitro Evaluation: Viability and Adhesion Cell

The samples were placed under ultraviolet (UV) light for 15 min for sterilization. The films were turned over so that the UV light irradiated both sides and they were completely sterilized.

For the proliferation viability assay, the samples were placed in 96-well plates and cultured with Primary Dermal Fibroblast Normal; Human, Neonatal (HDFn), (PCS-201-010, ATCC) with a concentration of 5 × 10^3^ cells/well and were maintained in an incubator at 37 °C with 5% CO_2_ in DMED medium with 10% fetal bovine (FBS) and 1% penicillin-streptomycin during 24 and 48 h. After each incubation time, the MTT medium was prepared as follows; for each 1 mL of medium, 5 µL of MTT reagent was added. The medium in the well-dish with cells was completely removed. Then, 200 µL of the MTT medium was added and incubated at an approximate temperature of 37 °C for one hour with 5% CO_2_. After this time, all possible MTT reagent was carefully removed after this time, and 200 µL of dimethyl sulfoxide was added. Finally, the well plates were placed in a UV/Visible spectrometer (Benchmark plus, BIO RAD, Monterrey, México) to measure the absorbance of the samples obtained at 570 nm. The MTT test data of the samples were compared with the optical density (OD) in the control group. The equation calculated the value of cell viability of the HDFn cells.
Cell viability (%) = A test/A control × 100 (2)
where A test is the OD of the test wells and A control—is the OD of the control wells. 

To prepare the acridine orange (AO) staining solution, 1 g of AO was dissolved in 100 mL of distilled water and stored away from light. Further, 3 × 10^4^ cells/well were seeded over 24 well culture dishes with the samples films and incubated at 37 °C with 5% CO_2_. After 24 and 48 h incubation, 50 µL of AO solution was added to each well, and the cells were observed on the optical microscopy. Images were captured and analyzed using an Axio vision Observer A.1 microscope (Zeiss, Jena, Germany).

#### 2.3.7. Statistical Analysis

The values shown in the results and discussion section have means ± standard deviation. When it is necessary, a *t* Student’s test was used to discern the statistical differences between results. A *p*-value of less than 0.05 was statistically significant.

## 3. Results and Discussion

### 3.1. Composite and Bi-Layer Films: Chemical Composition Study by FT-IR

Figure 2 shows the infrared spectra of the bilayer and composite films with different concentrations of gelatin. No significant differences were found between the six spectra of films since the signals are observed in all the film characteristics of both polymers. The IR signal observed at 3316 cm^−1^ represents intramolecular hydrogen bonds and the presence of NH bond stretching. In addition, a weak signal is found at 2921 cm^−1^ that is related to symmetric and asymmetric stretching of carbon bonds with hydrogen, which is associated with the characteristics of polysaccharides [12]. The 1650 cm^−1^ band corresponds to the presence of amide I, which represents the stretching of C = O. At 1555 cm^−1^, it indicates the presence of amide II which represents the bending vibration of the NH groups and the stretching of the CN groups. The 1240 cm^−1^ signal is associated with amide III, which, like amide II, represents the stretching vibration of the CN groups and the bending vibration of the NH groups. Likewise, the band at 1041 cm^−1^ is characteristic of pure chitosan and is associated with C-O stretching [13,14].

According to the previous investigations, the presence of both polymers in the bilayer and composite films observed in Figure 2 is confirmed since the essential characteristic signals of gelatin and chitosan were found, showing an optimal combination between these two polymers [14].

### 3.2. Thickness Measurement

The identification of the boundary between the gelatin and the chitosan of the bilayer films can be seen in Figure 3a–c, verifying that the bilayer films have a well-defined separation between the gelatin layer and the chitosan layer. Likewise, it is observed in Figure 3d–f that the composite films only consist of a uniform layer throughout the film, confirming the mixture of both polymers throughout the composite film. Using similar methods, different authors have obtained bilayer films of different polymers, among which bilayer films of alginate and chitosan with ciprofloxacin stand out as dressings for wound healing [15]. Further, bilayer films of chitosan and gelatin have been obtained with good antimicrobial activity [2] and desirable properties for food packaging [16].

From the results obtained and shown in Table 2, it can be observed that the difference in thickness in the bilayer films was lower compared to that of the composite films, so it can be concluded that the accommodation of the chains of both polymers is more homogeneous in bilayer films than in composite films. Particularly for composite films, this increase can be associated with a reduction in the alignment of the polymeric chains of each material due to the nature of the incorporated molecules, causing a reduction in the compaction of the network formed, which, in turn, increases the thickness of the films [17,18]. This behavior can be explained since, as the gelatin concentration increases, there is a higher solid content per surface unit, which causes an increase in thickness. In several investigations, this phenomenon was observed, where the increase in the thickness of the chitosan/montmorillonite films is attributed to a concentration effect since increasing the concentration of ginger essential oil increased the thickness of the films [19]. In addition, in another study, chitosan films were made with different concentrations of essential oils, from which a significant variation in the thickness of the films was obtained concerning the addition of these oils [17].

### 3.3. Surface Hydrophilicity by Contact Angle

Cell adhesion is the parameter to determine the biocompatibility of the biomaterials and can be studied with the contact angle of the surface. The contact angle of polymers is mainly attributed to the polymeric matrix’s immediate swelling and the material surface’s hygroscopicity. In the results shown in Figure 4 and Table 3, it is observed that, by having a higher concentration of gelatin in the layers of the analyzed films, the contact angle decreases compared to the layers of the films with less presence of gelatin. This is because chitosan chains can form ionic complexes from the anionic groups present, unlike gelatin, which only has a mostly cationic character [18]. An increase in the hydrophilic property of gelatin in relation to chitosan is associated with the thickness and amount of this protein deposited, which could absorb more water due to the presence of a high gamma of hydrophilic amino acids from gelatin [20].

In other investigations, it is observed that the presence of Ag coatings increases the contact angle of polymeric films [21]. At the same time, the influence of the morphology of the silver-coated surface can also be deduced since it can alter the film’s continuity by the coating, which can cause surface roughness, preventing the dispersion of the drop of silver liquid on the non-smooth surface (Figure 4b).

### 3.4. Degradation In Vitro Assay

In this study, an increase in the percentage of gradual degradation of all the films is observed over time (Figure 5). In addition, it is observed that the amount of gelatin and the configuration of the film helped the degradation of the films, because the films B2.5% and B3% had a degradation with a higher degradation on the last day of the study, with 64.16% and 74.19%, respectively. On the other hand, the film with the least degradation was C2%, with 46.12% on the third day. At the end of the degradation study, it was observed that the bilayer films obtained a higher percentage of degradation than the composite ones during the incubation time, with an average between the three films of 39%, 52%, and 62% for each day, and 27%, 47% and 53% for each day for composite films.

Chitosan is characterized by having great sensitivity to being degraded by different enzymes such as cellulases, pectinases, proteases, and lysozymes [22]. Lysozyme in the body protects against some microorganisms. Its primary action method is catalyzing the hydrolysis between the beta 1,4 bonds found in the residues of N-acetylmuramic acid and N-acetyl-D-glucosamine.

The degradation of films composed of different polymers has been studied in several investigations, such as films with O-carboxymethyl chitosan and gelatin with copper and silver-doped hydroxyapatite [5]. Where a similar degradation is observed at the end of the study, between 65 to 78% degradation of exposed films with the same enzyme, lysozyme [23]. Further, the films of O-carboxymethyl chitosan and extract of *Mimosa tenuiflora,* were exposed to the enzyme lysozyme, and a higher degradation was found at the end of the study of between 70 to 80% [24].

### 3.5. Antibacterial Properties Study by Inhibition Halo Assay

The zone of inhibition assay was used to determine the films’ antibacterial properties. The results are shown in Figure 6, and the measured diameters on Table 4. The indicator bacteria used in this test, *E. coli* and *S. aureus*, are common bacteria on the human skin [25]. The results showed that both bacteria exhibited sensitivity to Ag-coated films.

As can be seen in Figure 6, B2%, B2.5%, B3%, and C2% films that were exposed to the presence of *E. coli* bacteria have a higher inhibition halo, unlike the other films that only present an inhibition in the contact area, considering these last samples as bacteriostatic. This behavior can be attributed to the antibacterial properties of silver. Since they are films with a bilayer configuration, chitosan does not reduce their antimicrobial properties due to gelatin, unlike composite films. In similar analyses, Pereda et al. concluded that a possible reason for the action to be greater in those samples found in the agar impregnated with *E. coli* is due to the rupture of the lipopolysaccharide layer present in the outer membrane of Gram-negative bacteria [2]. Other studies have shown that Ag particles have a greater inhibition in gram-negative bacteria, such as *E. coli* [23].

Chitosan is widely known for its antimicrobial activity. It is attributed to the presence of positively charged amino groups, which can interact with negatively charged microbial cell membranes. It can cause leakage of proteins and other intracellular constituents, especially for Gram-negative bacteria [2]. This can be associated with the results obtained for bilayer films, where silver and the chitosan layer help to increase antimicrobial activity in the presence of *E. coli* bacteria.

Chitosan also has another possible mechanism of antimicrobial activity by inactivating the synthesis activity of the cell nucleus. In addition, the antimicrobial activity of chitosan in Gram-positive bacteria has been reported in different investigations [1]. This is observed in the case of *S. aureus* bacteria, where there is antimicrobial activity in both the bilayer and composite films, the latter being with a greater halo of inhibition. The results suggest that those film mats containing Ag-particles effectively control microbial growth on wound healing and can be functional in different in vivo systems studies.

### 3.6. Cell Viability Study

As in other investigations, fibroblasts were preferred on this assay as study cells because these are the first type of cells that meet the damaged tissue to have growth of tissue on the wound healing [23]. Figure 7 shows the results obtained from the cell viability percentage at 24 and 48 h of the bilayer and composite films at different gelatin concentrations. Taking as a reference the International Standard ISO 10993:2018-Annex C that indicates: If the viability is reduced to < 70% of the blank, it has a cytotoxic potential [26]. Furthermore, considering that the target is the control (well, with only cells, without the presence of any film), no film gave values less than 70%. Therefore, a cytotoxic characteristic is not shown.

The property of activating cell proliferation has been reported in various investigations for gelatin and chitosan. Previous studies attribute the proliferative property of gelatin to the presence of functional groups such as amino (-NH2) and hydroxyls (-OH) since they have shown an influence on the surface of a material in terms of cell adhesion, proliferation, and differentiation [27]. Furthermore, chitosan is a cationic polymer with positively charged amino groups, which interact with the negatively charged extracellular matrix, promoting cell adhesion and proliferation [27,28]. Films composed of chitosan and gelatin have been reported in various investigations to increase the cell viability of various cell lines, including murine fibroblasts [21] and adipose-derived stem cells [17].

Both the composite and bilayer films, coated with silver particles, did not turn out to be toxic. This behavior has been observed in other investigations, finding that, at low concentrations of silver in biomaterial coatings, cell capacity was maintained above 100% in fibroblastic cells, among which stand out: in chitosan gels [27], titanium implants [28], cotton fabric [25], and in cellulose nanofibers for wound healing [18]. It was shown that bilayer films would need a slightly higher cell capacity than composite films after 24 h of incubation, this can be explained by the contact angle results above, where bilayer films seemed to have a hydrophilic character greater than the composite ones, causing a greater adhesion of the cells in the first hours.

Cell adhesion was observed on all films when comparing the images in Figure 8 of the fibroblasts seeded on the bilayer and composite films. Additionally, fibroblasts in the proliferation phase were observed to a greater extent in the bilayer films, in addition to their morphological difference, being more elongated and with greater cellular interconnection than in the composite films, which, although they present cell adhesion, do not show a high interconnection and cell proliferation as in the case of bilayers. This may be because chitosan presents cell adhesion characteristics; however, gelatin has been reported in several investigations as a material that promotes cell proliferation and invasion, providing a greater hydrophilic character that allows cells to adhere and grow [5,28,29,30]. This confirms the results obtained from the hydrophilic characterization analysis, showing that the bilayer films show a greater hydrophilic character.

This proves that the cells seeded in the bilayer films have a more elongated morphology and a greater cell-cell interaction. In addition, improving cell adhesion and the confluence of the seeded fibroblasts, confirming the results obtained regarding cell viability using the previously performed MTT assay. Likewise, the results obtained confirm what was reported in previous investigation. Given each material’s optimal properties, when implementing a film with both polymers, it presents adhesion, migration, and cell growth [30].

## 4. Conclusions

Through the FTIR technique, observing the carbonyl, hydroxyl, and amide functional groups in the chemical structure of chitosan and gelatin present in the composite and bilayer films was possible due to the presence of the characteristic functional groups of gelatin and chitosan in all the films analyzed and, in addition, since the observed signals did not show considerable differences between them, the presence of both polymers was verified in all the films, giving, as a result, is an interaction between the two materials. According to the images obtained from the SEM, the composite films present a greater thickness, with 276 µm, 243.8 µm, and 239 µm compared to 236 µm, 233 µm, and 219 µm thick for bilayer films; and a low degradation rate compared to bilayer films. This is because the accommodation in the bilayer films occurs uniformly. While in the composite films, the gelatin and chitosan molecules collide with each other. In addition, the separation of chitosan and gelatin was obtained in the bilayer films, as observed in the SEM images.

The bilayer films exposed to the presence of Gram-negative bacteria demonstrated a greater perimeter in their inhibition halo than the composite ones, with an average of 0.89 cm. In turn, the absence of bacterial growth on the surface of all the films studied is notable. The bilayer films presented a higher percentage of degradation than the composite ones after 1, 2, and 3 days, with an average of 39%, 52%, and 62% for the bilayer films and 27%, 47%, and 53% for the composite films.

Finally, the MTT assay revealed no composite or bilayer films show cytotoxic characteristics. Likewise, a considerable difference was observed between the films with 24 and 48 h incubation, showing a higher percentage of viability at 48 h, proving to be a film-promoting proliferation. In the analyses of cell morphology, cell adhesion and proliferation were observed in all bilayer and composite films. The fibroblasts seeded on the films revealed cell-cell interaction and a characteristic elongated morphology. The bilayer films showed a more elongated morphological characteristic of the fibroblasts and a higher rate of cell proliferation and interaction, which corroborated the results obtained by MTT analysis. Therefore, chitosan/gelatin bilayer films with Ag coating can be used as dressings for wound healing due to their good antibacterial properties, cell viability in fibroblasts, and adequate degradation in vitro. However, it is recommended to carry out other characterizations to ensure its use, for example, mechanical studies and healing in vivo.

## Figures and Tables

**Figure 1 materials-16-03000-f001:**
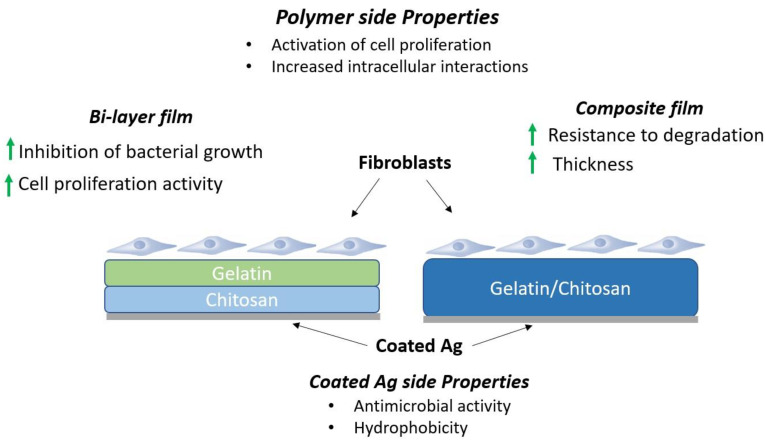
Graphic representation of gelatin and chitosan bi-layer and composite films with Ag coated.

**Figure 2 materials-16-03000-f002:**
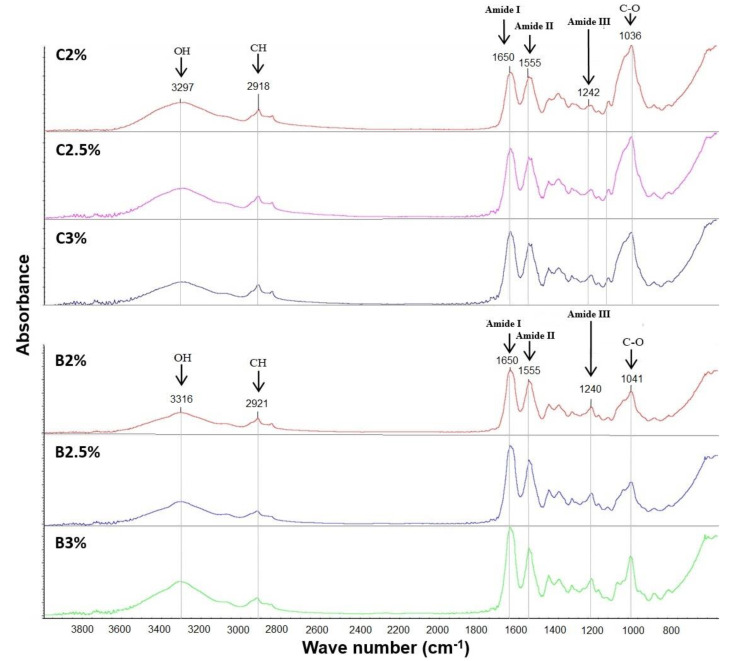
FTIR spectra of composite and bilayer films.

**Figure 3 materials-16-03000-f003:**
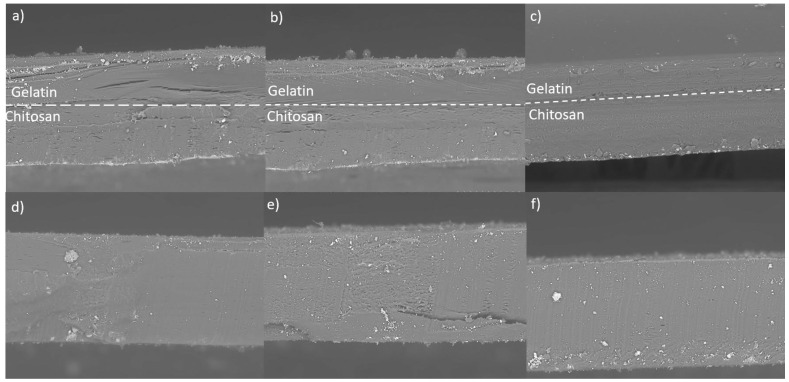
Observation of the thickness of the films in SEM: (**a**) 3% gelatin composite, (**b**) 2.5% gelatin composite, (**c**) 2% gelatin composite, (**d**) 3% gelatin bi-layer, (**e**) 2.5% gelatin bi-layer, (**f**) 2% gelatin bi-layer.

**Figure 4 materials-16-03000-f004:**
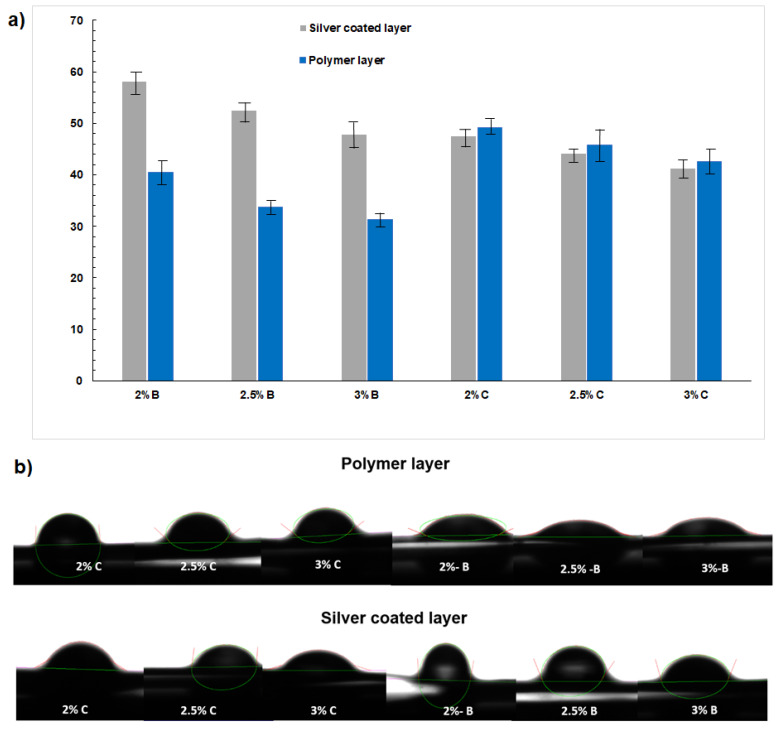
(**a**) Contact Angle Measurements of bilayer and composite Films, polymer layer, and silver Coating, (**b**) Images of the drop on the surfaces of the films.

**Figure 5 materials-16-03000-f005:**
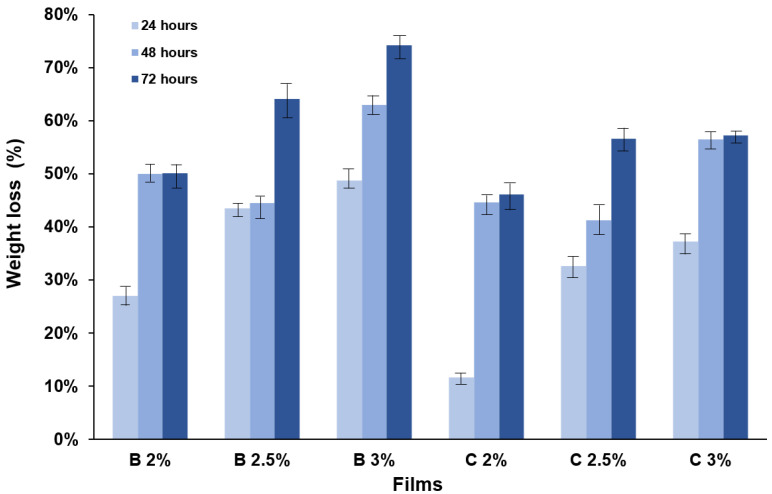
Percentage of weight loss (%WL) during enzymatic degradation on 24, 48, and 72 h of the composite and bilayer films.

**Figure 6 materials-16-03000-f006:**
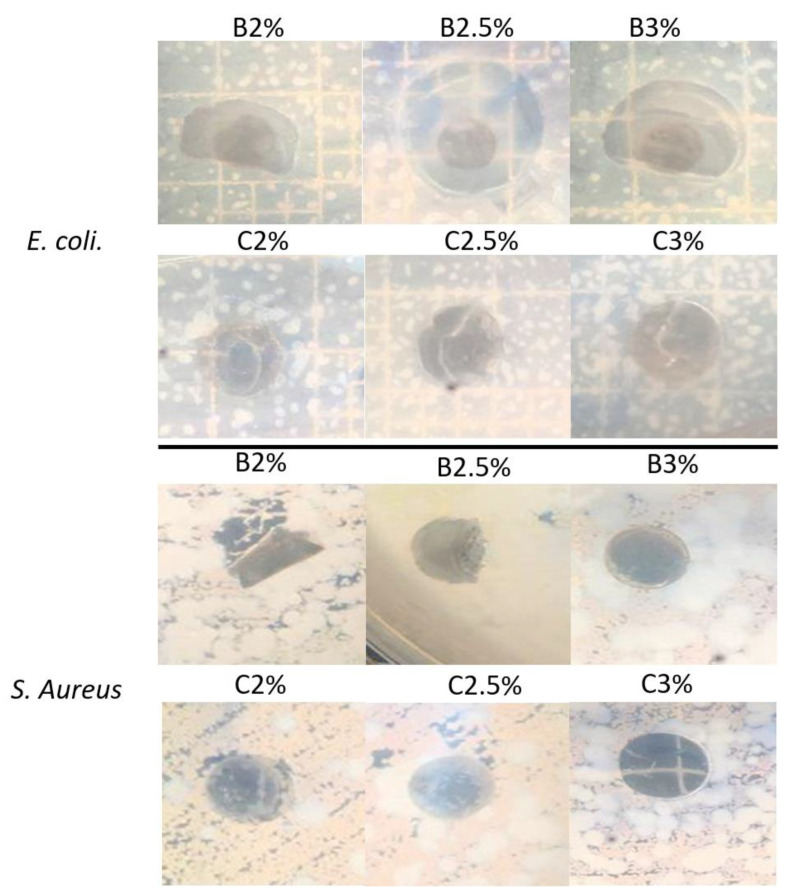
Halo test results of the bilayer and composite films with *E. coli* and *S. aureus* bacteria.

**Figure 7 materials-16-03000-f007:**
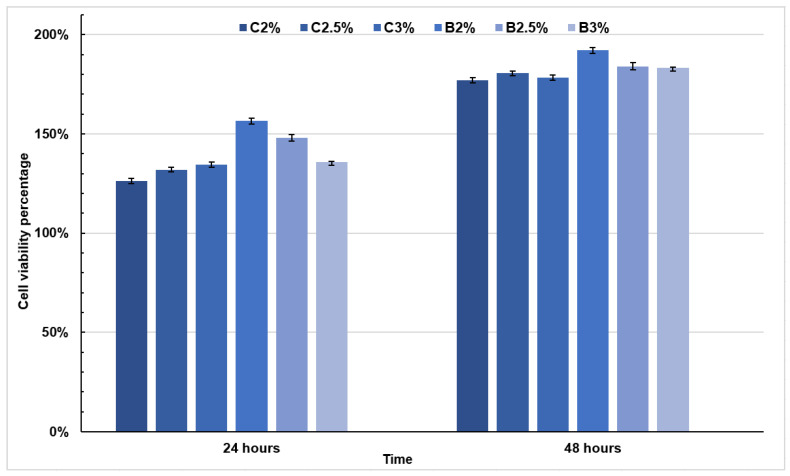
HDFn proliferation at 24 and 48 h grown on composite and bi-layer films, as assessed by MTT test.

**Figure 8 materials-16-03000-f008:**
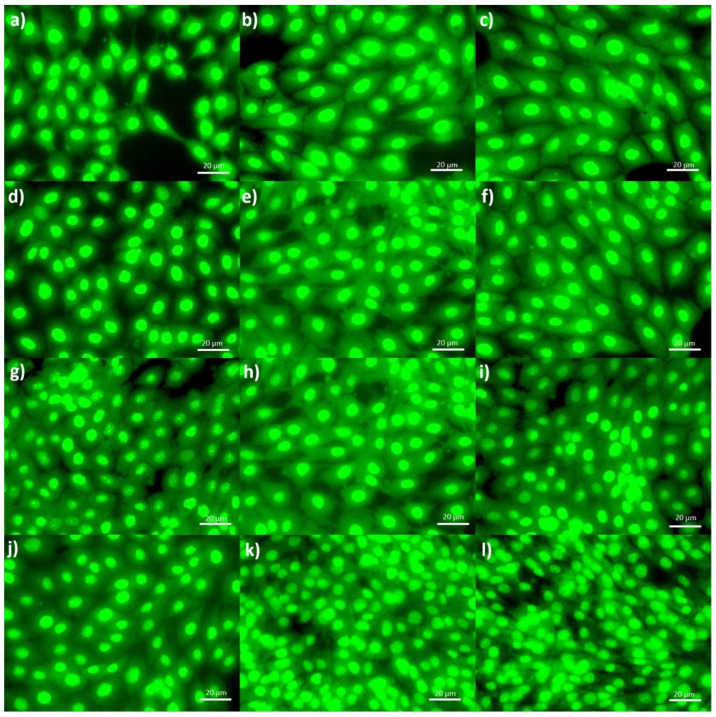
Morphology of HDFn cells stained with AO, grown for 24 (**a**–**f**) and for 48 h (**g**–**l**), on C2%, C2.5%, C3%, B2%, B2.5%, and B3% films, consecutively.

**Table 1 materials-16-03000-t001:** Film’s composition and coated time.

Film	% Chitosan	% Gelatin	Configuration	Coated Time with Ag
2% C	2%	2%	Composite	15 min
2.5% C	2%	2.5%	Composite	15 min
3% C	2%	3%	Composite	15 min
2% B	2%	2%	Bi-layer	15 min *
2.5% B	2%	2.5%	Bi-layer	15 min *
3% B	2%	3%	Bi-layer	15 min *

* The coats were realized on the chitosan layer.

**Table 2 materials-16-03000-t002:** Measurement of thickness films.

Film	Composite			Bi-Layer		
	2% C	2.5% C	3% C	2% B	2.5% B	3% B
Thickness (µm)	276 ± 2.25	243.8 ± 4.5	239 ± 4.1	236 ± 3.2	233 ± 2.4	219 ± 4.5

**Table 3 materials-16-03000-t003:** Contact angles of water for Ag coated and polymer layer.

Sample	Contact Angle (°)	
	Polymer Layer	Ag Coated Layer
2% C	49.2 a	47.5 a
2.5% C	45.8 b	44.1 b
3% C	42.5 c	41.2 c
2% B	40.4 d	58 e
2.5% B	33.7 f	52.4 g
3% B	31.3 h	47.8 i

Note: Two measurements in the same lines with 2 different letters are significantly different (*p* > 0.05) according to the *t* Student test.

**Table 4 materials-16-03000-t004:** Measurements of inhibition halos obtained by the agar diffusion method.

	*E. coli*		*S. aureus*
Composite films (inhibition halo mm)			
C2%	7.6 a		7.9 a
C2.5%	6.8 b		7.8 a
C3%	6.5 b		7.0 b
Bilayer films (inhibition halo mm)			
B2%	9.4 a		7.7 a
B2.5%	12.1 b		7.6 a
B3%	9.2 a		7.4 a

Note: Two measurements in the same column with 2 different letters are significantly different (*p* > 0.05) according to the *t* Student test.

## Data Availability

Not applicable.

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
