# Peer review of "Comparative Study of the Antibacterial, Biodegradable, and Biocompatibility Properties of Composite and Bi-Layer Films of Chitosan/Gelatin Coated with Silver Particles"

_materials, 2023, doi:10.3390/ma16083000_

Round 1

Reviewer 1 Report

Valencia-Gómez and colleagues and colleagues investigated a comparative study of the biocompatible, antibacterial, and biodegradable properties of the composite and bi-layer films, elaborated with chitosan and gelatin and coated with silver particles. The topic of this work is interesting. However, the manuscript must be improved. I would like to suggest a major revision and several issues should be addressed before consideration for publication. The suggestions are attached.

Author Response

Reviewer #1:

1.- Comment [M1]: You can use 1. traditional…(lowercase letter) 2….or a). traditional…; b)….; c…..

Before

The wound healing process has multi-factorial physiological activities and can be promoted with wound treatment or dressings. Currently, exists various dressings for heal skin, among which are: 1. Traditional dressings that are used in the first stage of therapy to stop bleeding and create a cover between the wound and environment; 2. Biomaterials-based dressings, biomimetic polymers with collagen-type structures; and 3. Artificial dressings, are the most effective materials in wound healing and have a longer shelf life [1].

After

The wound healing process has multi-factorial physiological activities and can be promoted with wound treatment or dressings. Currently, exists various dressings for healing skin, among which are: a) Traditional dressings that are used in the first stage of therapy to stop bleeding and create a cover between the wound and environment; b) Biomaterials-based dressings, biomimetic polymers with collagen-type structures; c) Artificial dressings, are the most effective materials in wound healing and have a longer shelf life [1].

2.- Font: (Default) Times New Roman, 10 pt, Not Bold.

Corrected.

3.- What is RGD? arginylglycylaspartic acid (RGD)

Before

such as RGD peptides,

After

such as arginine-glycine-aspartic motifs,

5.- You can enter a genconclusion after that.

Before

The objective of this investigation is to perform a comparative study of the biocompatible, antibacterial, and biodegradable properties of the composite and bi-layer films, elaborated with chitosan and gelatin and coated with silver particles, to determine the most efficient configuration for dressings on wound healing applications (Figure 1).

After

The objective of this investigation is to perform a comparative study of the biocompatible, antibacterial, and biodegradable properties of the composite and bi-layer films, elaborated with chitosan and gelatin and coated with silver particles, to determine the most efficient configuration for dressings on wound healing applications, through from the study of cell viability in vitro, enzymatic degradation in vitro, the halo of inhibition and contact angle of the elaborated films (Figure 1).  This comparative study will help to understand the effect of the configuration of materials on the properties of chitosan/gelatin films. In addition, this research will provide more information on the use of bilayer films in wound care applications due to the limited current information on this type of film for medical applications.

6.- Where are the materials?

Before

  • Materials

After

  • Materials

The materials and solvents used to elaborate the films were chitosan from shrimp shells ≥ 75% deacetylated (Sigma-Aldrich, C3646), gelatin from bovine skin (Sigma-Aldrich, G9382), acetic acid (T.Baker, 9508-05). For the characterizations and tests, the solvents were Dimethyl sulfoxide (DMSO) (Sigma-Aldrich, D4540), 3-(4,5-Dimethyl-2-thiazolyl)-2,5-diphenyl-2H-tetrazolium Bromide or MTT (Sigma-Aldrich, 475989), 1x DMEM culture medium (Gibco), 10% Fetal Bovine Serum (ATCC), 2% penicillin (Gibco), Accutase enzyme (Sigma Aldrich), Soy Agar (Difco, 236950), Nutrient Agar (Difco, 21300), and lysozyme enzyme (Fisher Scientific, BP535-1).

7.- %

After

The composite films were prepared using a modified methodology reported by Valencia-Gomez, in 2022 [5]. Three different gelatin solutions were prepared with different concentrations, 2%, 2.5% and 3%,

8.-  English check please

The manuscript was revised and spelling mistakes were corrected (Grammarly was used).

9.- Percentage of weight loss (%WL)

Before

Wl%

After

 %WL

10.- The equations show be numbered.

Before

Wl%= Wi -  Wf   X 100%

Wi

After

                                 %WL= Wi -  Wf   X 100%                          (1)

Wi

                          Cell viability (%) = A test/A control × 100                       (2)

11.- Font: (Default) Times New Roman, 10 pt, Not Bold

Before

concentration of 5X103 cells/well

After

with a concentration of 5 X 103 cells/well

12.- Space after 37

Before

temperature of 37°C

After

temperature of 37 °C for

13.- You can use once Absorbance like Wave number

Before

After

14.- Here is point.

Before

[16]; In addition,

After

[20]. In addition,

15.- The figure may be smaller

The figure 7 became smaller.

16.- Some references are in bold, others are not.

The writing of references was homogenized.

17.- What is plotted on x-axis? the figure may be smaller

The figure 7 became smaller.

18.- Conclusions could limited.

Before

Through the FTIR technique, it was possible to observe the carbonyl, hydroxyl, and amide functional groups that are part of the chemical structure of chitosan and gelatin, present in the composite and bilayer films. Due to the presence of the characteristic functional groups of gelatin and chitosan in all the films analyzed and, in addition, since the observed signals did not show considerable differences between them, the presence of both polymers was verified in all the films, giving as the result is an interaction between the two materials. According to the images obtained from the SEM, the composite films present a greater thickness than the bilayer ones, this is because the accommodation in the bilayer films occurs uniformly while in the composite films, the gelatin and chitosan molecules collide with each other. In addition, the separation of chitosan and gelatin was obtained in the bilayer films, as observed in the SEM images.

The bilayer films exposed to the presence of Gram-negative bacteria demonstrated a greater perimeter in their inhibition halo than the composite ones with an average of 0.89 cm. In turn, the absence of bacterial growth on the surface of all the films studied is notable. The bilayer films presented a higher percentage of degradation than the composite ones after 1, 2 and, 3 days, with an average of 39%, 52% ,and 62% for the bilayer films and 27%, 47%, and 53% for the composite ones.

Finally, the MTT assay revealed that no composite or bilayer films show cytotoxic characteristics. Likewise, a considerable difference was observed between the films with 24 and 48 h of incubation, showing a higher percentage of viability at 48 h, proving to be a film that promotes proliferation.

In the analyses of cell morphology, cell adhesion, and proliferation were observed in all bilayer and composite films. The fibroblasts seeded on the films revealed cell-cell interaction and a characteristic elongated morphology. The bilayer films showed a more elongated morphological characteristic of the fibroblasts, in addition to a higher rate of cell proliferation and interaction, which corroborated the results obtained by MTT analysis.

After

Through the FTIR technique, observing the carbonyl, hydroxyl, and amide functional groups in the chemical structure of chitosan and gelatin present in the composite and bilayer films was possible Due to the presence of the characteristic functional groups of gelatin and chitosan in all the films analyzed and, in addition, since the observed signals did not show considerable differences between them, the presence of both polymers was verified in all the films, giving, as a result, is an interaction between the two materials. According to the images obtained from the SEM, the composite films present a greater thickness, with 276 µm, 243.8 µm, and 239 µm compared to 236 µm, 233 µm and 219 µm thick for bilayer films; and a low degradation rate compared to bilayer films. This is because the accommodation in the bilayer films occurs uniformly. while in the composite films, the gelatin and chitosan molecules collide with each other. In addition, the separation of chitosan and gelatin was obtained in the bilayer films, as observed in the SEM images.

The bilayer films exposed to the presence of Gram-negative bacteria demonstrated a greater perimeter in their inhibition halo than the composite ones, with an average of 0.89 cm. In turn, the absence of bacterial growth on the surface of all the films studied is notable. The bilayer films presented a higher percentage of degradation than the composite ones after 1, 2. and 3 days, with an average of 39%, 52%, and 62% for the bilayer films and 27%, 47%, and 53% for the composite films.

Finally, the MTT assay revealed no composite or bilayer films show cytotoxic characteristics. Likewise, a considerable difference was observed between the films with 24 and 48 h incubation, showing a higher percentage of viability at 48 h, proving to be a film-promoting proliferation. In the analyses of cell morphology, cell adhesion and proliferation were observed in all bilayer and composite films. The fibroblasts seeded on the films revealed cell-cell interaction and a characteristic elongated morphology. The bilayer films showed a more elongated morphological characteristic of the fibroblasts and a higher rate of cell proliferation and interaction, which corroborated the results obtained by MTT analysis. Therefore, chitosan/gelatin bilayer films with Ag coating can be used as dressings for wound healing due to their good antibacterial properties, cell viability in fibroblasts, and adequate degradation in vitro However, it is recommended to carry out other characterizations to ensure its use, for example, mechanical studies and healing in vivo.

Reviewer 2 Report

Incorporate suggestions!

Author Response

Reviewer #2:

1.- . Abstract of the article is not suitable. More theoretical information is provided whereas quantitative values are required. Revised the abstract properly.

Before

The dressings are materials that can improve the wound-healing process in patients with medical issues. Polymeric films are frequently used as dressings with multiple biological properties. Chitosan and gelatin are some of the most used polymers in tissue regeneration processes.  Usually, there are several configurations of films for dressings, among which the composite (mixture of two or more materials) and layered ones stand out (layers). In this study, the antibacterial, degradable, and biocompatible properties of chitosan and gelatin films in 2 configurations, composite, and bilayer, were analyzed. In addition, a silver coating was added to enhance the antibacterial properties of both configurations. After the study, it was found that bilayer films have greater antibacterial activity than composite films and increase the process of fibroblast cell proliferation. On the other hand, composite films have greater stability since they have a greater thickness and a low rate of degradation compared to bilayer films.

After

The dressings are materials that can improve the wound-healing process in patients with medical issues. Polymeric films are frequently used as dressings with multiple biological properties. Chitosan and gelatin are the most used polymers in tissue regeneration processes.  Usually, there are several configurations of films for dressings, among which the composite (mixture of two or more materials) and layered ones stand out (layers). This study analyzed the antibacterial, degradable, and biocompatible properties of chitosan and gelatin films in 2 configurations, composite and bilayer, composite. In addition, a silver coating was added to enhance the antibacterial properties of both configurations. After the study, it was found that the bilayer films have a higher antibacterial activity than the composite films, having inhibition halos between 23% and 78% in Gram-negative bacteria. In addition, the bilayer films increased the fibroblast cell proliferation process, reaching up to 192% cell viability after 48 hours of incubation. On the other hand, composite films have greater stability since they are thicker, with 276 µm, 243.8 µm and 239 µm compared to 236 µm, 233 µm, and 219 µm thick for bilayer films; and a low degradation rate compared to bilayer films.

2.- What edge does this paper have over the existing works? This needs to be presented as a novelty statement in the introduction.

Before

The objective of this investigation is to perform a comparative study of the biocompatible, antibacterial, and biodegradable properties of the composite and bi-layer films, elaborated with chitosan and gelatin and coated with silver particles, to determine the most efficient configuration for dressings on wound healing applications (Figure 1).

After

The objective of this investigation is to perform a comparative study of the biocompatible, antibacterial, and biodegradable properties of the composite and bi-layer films, elaborated with chitosan and gelatin and coated with silver particles, to determine the most efficient configuration for dressings on wound healing applications, through from the study of cell viability in vitro, enzymatic degradation in vitro, the halo of inhibition and contact angle of the elaborated films (Figure 1).  This comparative study will help to understand the effect of the configuration of materials on the properties of chitosan/gelatin films. In addition, this research will provide more information on the use of bilayer films in wound care applications due to the limited current information on this type of film for medical applications.

3.- Introduction is written simply; most recent research and innovation on this work must be added to align the work as latest research.

Before

Silver particles are used in skin lesion dressings, given their low level of toxicity to human cells, easy access, and strong antimicrobial effect [6]. Dressings with antimicrobial properties are useful in protecting the wound from infection and eliminating pain. Silver is an antimicrobial agent that has shown activity against more common pathogens such as Staphylococcus aureus, Escherichia coli, and Pseudomonas spp [7]. In addition, this material has previously been used as an antimicrobial agent for biomedical applications, adding it using the sputtering method, which has been reported as a technique that allows depositing a thin layer of silver, which, depending on its deposition time, may remain with the characteristic antimicrobial character of this material [8].

The objective of this investigation is to perform a comparative study of the biocompatible, antibacterial, and biodegradable properties of the composite and bi-layer films, elaborated with chitosan and gelatin and coated with silver

After

Silver particles are used in skin lesion dressings, given their low level of toxicity to human cells, easy access, and strong antimicrobial effect [6]. Dressings with antimicrobial properties protect the wound from infection and eliminate pain. Silver is an antimicrobial agent that has shown activity against more common pathogens such as Staphylococcus aureus, Escherichia coli, and Pseudomonas spp [7]. In addition, this material has previously been used as an antimicrobial agent for biomedical applications, adding it using the sputtering method, which has been reported as a technique that allows depositing a thin layer of silver, which, depending on its deposition time, may remain with the characteristic antimicrobial character of this material [8].

Various recent investigations have been carried out on manufacturing chitosan-based bilayer films. However, these studies are focused on the preparation of food packaging, for example, chitosan–poly (vinyl alcohol) bilayer films for Strawberry packaging [9] and modified chitosan/chitosan nanoparticle and polyvinyl alcohol/starch bilayer films [10]. However, there is limited research on chitosan-based bilayer films for wound healing applications. For example, chitosan/konjac glucomannan bilayer film as a wound dressing, where it is observed that bilayer films have low cytotoxicity and inhibit microbial penetration [11]. Also, in 2019, gelatin/chitosan bilayer hydrofilms with crosslinking agents were studied, showing good properties for wound dressing use [12]. Nevertheless, chitosan/gelatin films coated Ag have not been studied for wound healing applications.

The objective of this investigation is to perform a comparative study of the biocompatible, antibacterial, and biodegradable properties of the composit

3.- The figure 1 name should consist of the materials name also.

Before

Figure 1. Graphic representation of the bi-layer and composite films with Ag coated.

After

Figure 1. Graphic representation of gelatin and chitosan  bi-layer and composite films with Ag coated.

4.- In section 2. 1.. nothing is mentioned please check.

After

  • Materials

The materials and solvents used to elaborated the films were chitosan from shrimp shells ≥ 75% deacetylated (Sigma-Aldrich, C3646), gelatin from bovine skin (Sigma-Aldrich, G9382), acetic acid (T.Baker, 9508-05). For the characterizations and tests, the solvents were Dimethyl sulfoxide (DMSO) (Sigma-Aldrich, D4540), 3-(4,5-Dimethyl-2-thiazolyl)-2,5-diphenyl-2H-tetrazolium Bromide or MTT (Sigma-Aldrich, 475989), 1x DMEM culture medium (Gibco), 10% Fetal Bovine Serum (ATCC), 2% penicillin (Gibco), Accutase enzyme (Sigma Aldrich), Soy Agar (Difco, 236950), Nutrient Agar (Difco, 21300), and lysozyme enzyme (Fisher Scientific, BP535-1).

5.-  Mechanical studies also play a vital role. It is suggested to study some of the mechanical behaviors of such composites and bilayer composites.

Mechanical tests were not performed on the films; however, its recommendation is mentioned in the conclusions.

6.- The figure 2 is required to replot again as it will not clear when adjusted in the format of paper.

Corrected

7.- In section 3.3, The contact angle value of the studied composition should also be in table form. please check.

After

Table 3. Contact angles of water for Ag coated and polymer layer.

Sample

Contact Angle (°)

Polymer layer

Ag coated layer

2%C

49.2 a

47.5a

2.5%C

45.8 b

44.1 b

3%C

42.5 c

41.2 c

2%B

40.4 d

58 e

2.5%B

33.7 f

52.4 g

3%B

31.3 h

47.8 i

Note: Two measurements in the same lines with 2 different letters are significantly different (P>0.05) according to the T Student test.

8.- In figure 6, figure no. and scale bar are missing.

The measurement of the inhibition halos is found in the table 4.

9.- In sentence no.318,what is table 26.Please check

Checked.

10.- In conclusion section, future of this work must be aligned in revised article.

Before

Through the FTIR technique, it was possible to observe the carbonyl, hydroxyl, and amide functional groups that are part of the chemical structure of chitosan and gelatin, present in the composite and bilayer films. Due to the presence of the characteristic functional groups of gelatin and chitosan in all the films analyzed and, in addition, since the observed signals did not show considerable differences between them, the presence of both polymers was verified in all the films, giving as the result is an interaction between the two materials. According to the images obtained from the SEM, the composite films present a greater thickness than the bilayer ones, this is because the accommodation in the bilayer films occurs uniformly while in the composite films, the gelatin and chitosan molecules collide with each other. In addition, the separation of chitosan and gelatin was obtained in the bilayer films, as observed in the SEM images.

The bilayer films exposed to the presence of Gram-negative bacteria demonstrated a greater perimeter in their inhibition halo than the composite ones with an average of 0.89 cm. In turn, the absence of bacterial growth on the surface of all the films studied is notable. The bilayer films presented a higher percentage of degradation than the composite ones after 1, 2 and, 3 days, with an average of 39%, 52% ,and 62% for the bilayer films and 27%, 47%, and 53% for the composite ones.

Finally, the MTT assay revealed that no composite or bilayer films show cytotoxic characteristics. Likewise, a considerable difference was observed between the films with 24 and 48 h of incubation, showing a higher percentage of viability at 48 h, proving to be a film that promotes proliferation.

In the analyses of cell morphology, cell adhesion, and proliferation were observed in all bilayer and composite films. The fibroblasts seeded on the films revealed cell-cell interaction and a characteristic elongated morphology. The bilayer films showed a more elongated morphological characteristic of the fibroblasts, in addition to a higher rate of cell proliferation and interaction, which corroborated the results obtained by MTT analysis.

After

Through the FTIR technique, observing the carbonyl, hydroxyl, and amide functional groups in the chemical structure of chitosan and gelatin present in the composite and bilayer films was possible Due to the presence of the characteristic functional groups of gelatin and chitosan in all the films analyzed and, in addition, since the observed signals did not show considerable differences between them, the presence of both polymers was verified in all the films, giving, as a result, is an interaction between the two materials. According to the images obtained from the SEM, the composite films present a greater thickness, with 276 µm, 243.8 µm, and 239 µm compared to 236 µm, 233 µm and 219 µm thick for bilayer films; and a low degradation rate compared to bilayer films. This is because the accommodation in the bilayer films occurs uniformly. while in the composite films, the gelatin and chitosan molecules collide with each other. In addition, the separation of chitosan and gelatin was obtained in the bilayer films, as observed in the SEM images.

The bilayer films exposed to the presence of Gram-negative bacteria demonstrated a greater perimeter in their inhibition halo than the composite ones, with an average of 0.89 cm. In turn, the absence of bacterial growth on the surface of all the films studied is notable. The bilayer films presented a higher percentage of degradation than the composite ones after 1, 2. and 3 days, with an average of 39%, 52%, and 62% for the bilayer films and 27%, 47%, and 53% for the composite films.

Finally, the MTT assay revealed no composite or bilayer films show cytotoxic characteristics. Likewise, a considerable difference was observed between the films with 24 and 48 h incubation, showing a higher percentage of viability at 48 h, proving to be a film-promoting proliferation. In the analyses of cell morphology, cell adhesion and proliferation were observed in all bilayer and composite films. The fibroblasts seeded on the films revealed cell-cell interaction and a characteristic elongated morphology. The bilayer films showed a more elongated morphological characteristic of the fibroblasts and a higher rate of cell proliferation and interaction, which corroborated the results obtained by MTT analysis. Therefore, chitosan/gelatin bilayer films with Ag coating can be used as dressings for wound healing due to their good antibacterial properties, cell viability in fibroblasts, and adequate degradation in vitro However, it is recommended to carry out other characterizations to ensure its use, for example, mechanical studies and healing in vivo.

8.- Pay close attention to style guidelines (formatting for references/citations in text and equation formatting).

Checked.

9.- The figure 8 is not aligned properly. Red color indication is suggested to change to some other color.

After

10.- In figure 4 a, what was the value of the std. deviation, it should be mentioned in tabular form. Similarly for figure 5, it should be done.

After

11.-  In sentence no.318,what is table 26.Please check “Table 26” works are removed.

12.- Highlight the future scope of present research work. It was added in the conclusion section.

13.-  Pay close attention to style guidelines (formatting for references/citations in text and equation formatting).

Reference format changed. 

14.- The figure 2 is required to replot again as it will not clear when adjusted in

the format of paper.  Figure 2 was modified.

Reviewer 3 Report

Comments:
The manuscript reports on the “Comparative study of the antibacterial, biodegradable, and biocompatibility properties of composite and bilayer films of chitosan/gelatin coated with silver particles”. The topic is timely and important, but the manuscript has several shortcomings that preclude its acceptance for publication.

1.      The Turnitin report attached to the manuscript indicates a need for further work to reduce plagiarism and provide more justification for the results and novelty of the work presented. The authors should carefully review the manuscript and ensure that all sources are appropriately cited and any passages that may be deemed problematic are rephrased.

2.      What technique was used to observe the functional groups present in chitosan and gelatin in the composite and bilayer films?

3.      What was the difference in thickness between composite and bilayer films, and why did it occur?

4.      Please mention that Is the bacterial growth observed on the surface of any of the films.

5.      Have you checked the cytotoxic of the composite and bilayer films?

6.      What was the result of the MTT assay, and how did it vary with time?

7.      How did the fibroblasts seeded on the films interact with each other and what was their characteristic morphology?

8.      Did the bilayer or composite films show a higher rate of cell proliferation and interaction, and how did this compare to the MTT analysis?

Author Response

Reviewer #3:

1.- The Turnitin report attached to the manuscript indicates a need for further work to reduce plagiarism and provide more justification for the results and novelty of the work presented. The authors should carefully review the manuscript and ensure that all sources are appropriately cited and any passages that may be deemed problematic are rephrased.

Reviewed.

  1. What techniquewas used to observe the functional groups present in chitosan and gelatin in the composite and bilayer films? The technique used was Fourier transform infrared spectroscopy, the results of the gelatin and chitosan groups can be seen in Figure 2, where the spectra of the composite and bilayer films are observed.
  2. What was the difference in thickness between composite and bilayer films, and why did it occur? It was observed that the composite films had a greater thickness than the bilayer films, this is because the bilayer films have a better ordering in the polymer chains compared to the composite films, which were manufactured by mixing both polymers in a solution.
  3. Please mention that Is the bacterial growth observed on the surface of any of the films. Bacterial growth was not observed in the films, however, a significant difference was found in E. coli bacteria, where the bilayer films had better antibacterial activity.
  4. Have you checked the cytotoxicof the composite and bilayer films? The cytotoxicity of both films in fibroblasts was verified, where it was verified that they are not toxic, in addition to having good cell viability during the two days of incubation.
  5. What was the result of the MTT assay, and how did it vary with time? The MTT results can be seen in Figure 7, where it is observed that cell viability increases, confirming that the incubation time of both films passes.
  6. How did the fibroblasts seeded on the films interact with each other and what was their characteristic morphology? The fibroblasts seeded on the films were observed by staining, and the images are shown in Figure 8. In the microscope images the connections between cells are identified, which indicates that the cells begin to create a tissue on the films.
  7. Did the bilayer or composite films show a higher rate of cell proliferationand interaction, and how did this compare to the MTT analysis? Both films showed cell proliferation in the seeded fibroblasts. This cell growth can be seen in the MTT results, in Figure 7, as cell viability increases, suggesting that the cells pro-released during the incubation time. It is also observed in Figure 8, where the number of cells increases on the surface of the films.

Round 2

Reviewer 1 Report

Thanks to the authors for the revision, the manuscript can be accepted in the form presented.